# Automatic Identification of Sedimentary Facies Based on a Support Vector Machine in the Aryskum Graben, Kazakhstan

**Xiao Ai** [1] , **Hongyu Wang** [2] **and Baitao Sun** [1,*]

1   Institute of Engineering Mechanics, China Earthquake Administration. Key Laboratory of Earthquake
    Engineering and Engineering Vibration, China Earthquake Administration, Harbin 150080, China;
    axz.1988@163.com
2   School of Energy Resources, China University of Geosciences, Beijing 10083, China; wanghy@cugb.edu.cn
*   Correspondence: sunbt@iem.cn

**Abstract:** The Aryskum Depression in the South Turgay Basin has shown improving exploration prospects for subtle reservoirs, due to investment in the exploration workload and more comprehensive geological research. Among them, lithologic stratigraphic reservoirs have gradually become one of the focuses of oil and gas exploration. At present, deduction of the sedimentary characteristics of the target layer through core wells using artificial exploration has become an urgent problem to be solved. We selected 16 artificially interpreted coring wells in the Aryskum Graben for this study. Using the parameters of the gamma-ray (GR) curve of coring wells and support vector machine (SVM) classification algorithms, we developed an automatic identification model of sedimentary facies in the study area. The application of the SVM includes the following steps: Firstly, using the GR curve of 16 coring wells, six quantitative indexes defined as standard deviation, relative gravity, curve amplitude ratio, average median, average slope, and mutation amplitude, are selected to quantify the logging curve in the study area, thus realizing the description of the logging curve form. Secondly, training samples are selected to establish an SVM classification model. Finally, a quantitative discrimination model based on the SVM algorithm is established to realize the classification of depositional facies. Field application shows that this solution can be effectively used in uncored wells to identify depositional facies with a rate of accuracy approaching 70%. Our results provide new methods for the identification of sedimentary facies in the study area. The results will also provide a theoretical basis, as well as data basis, for further fine division of microfacies in the study area.

**Keywords:** machine learning; support vector machine; South Turgay Basin; Aryskum Graben; depositional subfacies; automatic identification model; data mining

---

## 1. Introduction

The oil and gas discovered in the South Turgay Basin are mainly located in the Aryskum Depression in the south, across an area of about $3 \times 10^4$ km$^2$. The main sedimentary strata in the Aryskum Depression are Jurassic–Cretaceous fluvial sandstone and mudstone, the main source rocks are Middle–Lower Jurassic dark mudstone, and the main reservoir rocks are Mesozoic sandstone and buried hill. The entire basin contains four sets of reservoir-forming assemblages: $K_1$, $J_3$, $J_{1-2}$, and $P_z$. At present, the main development strata are $J_3$ and $K_1$. Since the basin has undergone more than 40 years of exploration, the basic drilling of large-scale structural traps is nearly at an end. Therefore, searching for stratigraphic lithologic traps has become a new target for increasing reserves. Particularly in recent years, the Aryskum Depression has shown improving exploration prospects for subtle reservoirs, due to investment in the exploration workload and more comprehensive geological

research, and lithologic stratigraphic reservoirs have gradually become some of the main focuses of oil and gas exploration.

The determination of sedimentary facies plays an important role in the exploration of lithologic reservoirs, especially in the prediction of residual oil production. In addition, coring and logging are seldom carried out in development wells. Therefore, increasing attention has been paid to the method of studying sedimentary facies by logging curves. However, the process of manual identification often incurs many problems, such as an intense workload of data statistics and mapping, low work efficiency, poor quantification, and non-uniformity of the criteria for manual identification of sedimentary facies in uncored wells. These problems highlight that there are many shortcomings in the artificial identification of multi-well and multi-layer logging sedimentary facies. With the popularization of digital logging technology, digital logging curve technology, and computer technology, the machine learning method in particular has the advantages of dealing with non-linear mapping relations. In recent years, machine learning has been preliminarily applied in processing logging data and predicting reservoir pore and permeability properties.

International geologists are currently conducting research on the identification of sedimentary facies by machine learning. This mainly includes the KNN (k-nearest neighbor) [1], ANN (artificial neural network) [2–5], Bayesian [6], fuzzy clustering [7,8], and support vector machine (SVM) algorithms [9–11]. However, some of these machine learning algorithms have limitations and, in many cases, may not meet the requirements for the identification of sedimentary facies [12,13]. KNN requires a large amount of computation and there is an issue of sample imbalance; that is, the number of samples in some categories is large, while the number in other categories is very small. Although the artificial neural network algorithm has a strong ability to deal with the problem of non-linear classification, it is easy to cause model over-fitting due to its strong non-linear ability, and it requires a large number of parameters. Besides, as the learning process cannot be observed, the output is difficult to understand [14]. When using a Bayesian algorithm, it is necessary to assume that the experimental samples are distributed independently, and the assumptions of the independent distribution should be known first [15]. Fuzzy clustering analysis is sensitive to noise and outliers. Additionally, its results are unstable [16].

The theory of SVM was first proposed by Vapnik in 1995 [17]. SVM is a machine learning algorithm that relies on statistical theory and Vapnik–Chervonenkis (VC) dimension theory, and is based on the structural risk minimization principle, which is to seek the best balance between the complexity of the model and the learning ability of the model through limited sample information. Compared with an artificial neural network, it has a stronger generalization ability. The data of this study are different from those commonly used in machine learning. Logging data have the characteristics of fewer samples and unbalanced numbers of samples. To solve this problem, an SVM algorithm has unique advantages in small- and medium-sized sample statistical processing and is suitable for classification tasks with unbalanced numbers of samples [18,19]. Moreover, an SVM is less prone to over-fitting, and has the significant advantage of being without a local minimum [20,21].

## 2. Geological Setting

### 2.1. Regional Geological Conditions

The South Turgay Basin is one of the main oil-producing basins in Kazakhstan. It is located in the central part of Kazakhstan and belongs to the Mesozoic rift basin above the Hercynian basement. Its evolution has experienced four stages: basement formation, early Mesozoic rift, late Mesozoic post-rift, and Himalayan compressive collision. The general trend of the basin is from northwest to southeast and it has an area of about $8 \times 10^4$ km$^2$. The northern part of the South Turgay Basin is the North Turgay Basin, the western part is the Ural Suture Belt, and the eastern part is the Ulutau Uplift. The Aryskum Depression is located in the southern part of the South Turgay Basin. The strike–slip fault of Karatau has the same strike–slip trend as that of the South Turgay Basin, and the

strike–slip fault is northwest–southeast. It is connected with the Northwest Ural orogenic belt and extends southeast through the Kashi Depression of the South Turgay Basin and Tarim Basin [22,23]. The Aryskum Depression has a horst and graben geological structure in plane and can be subdivided into four grabens and three horsts in a sequence from west to east of: Aryskum Graben, Aksay Horst, Akshabulak Graben, Ashisay Horst, Sarylan Graben, Tabakbulak Horst, and Bozingen Graben. The study area is the Aryskum Graben, located in the western part of the Aryskum Depression [24]. Drilling data in the study area are abundant, with more than 90 wells, which are distributed across the whole Aryskum Graben area. In this study, 16 artificially interpreted coring wells in the study area were selected as experimental data and named ARY-01–ARY-16 (Figure 1).

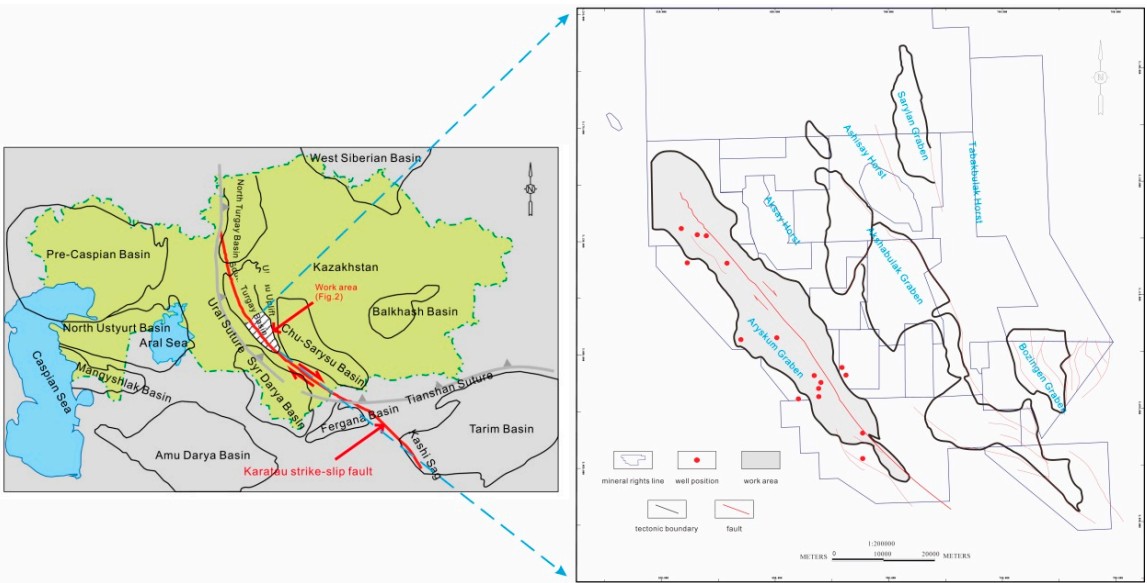

**Figure 1.** Location of work area and distribution of coring wells.

## 2.2. Character of Strata Development

The South Turgay Basin developed an early Mesozoic basement and upper unconformity sedimentary layers, which are a Jurassic rift system and Cretaceous quaternary post-fracture depression sedimentary system, respectively [25] (Figure 2).

In the Early–Middle Jurassic, lacustrine mudstone was the main layer type. The lateral margin of the graben is sometimes found to be transformed into coarse-grained gravel and estuarine clastic rocks. The estuarine deposits are widely distributed, and the provenance comes from the basement uplift of the lateral margin of the graben.

During the Jurassic–Cretaceous transition period, the rift deposits occured. Grabens ceased to move strongly, the whole basin turned into a slow depression, and parts of the basin suffered denudation, forming the bottom sandstone of the Cretaceous, that is, the Aryskum stratum. From Figure 3, it can be seen that the Aryskum Graben sedimentary facies are generally fluvial and delta facies in the Late Jurassic Period. The delta sediments developed extensively during the filling period, and fluvial facies were well developed in the northern part of the South Turgay Basin. During this period, the river area increased, and the lake became shallow and moved southward. Semi-deep lake sediments can be seen in the Bozingen Graben.

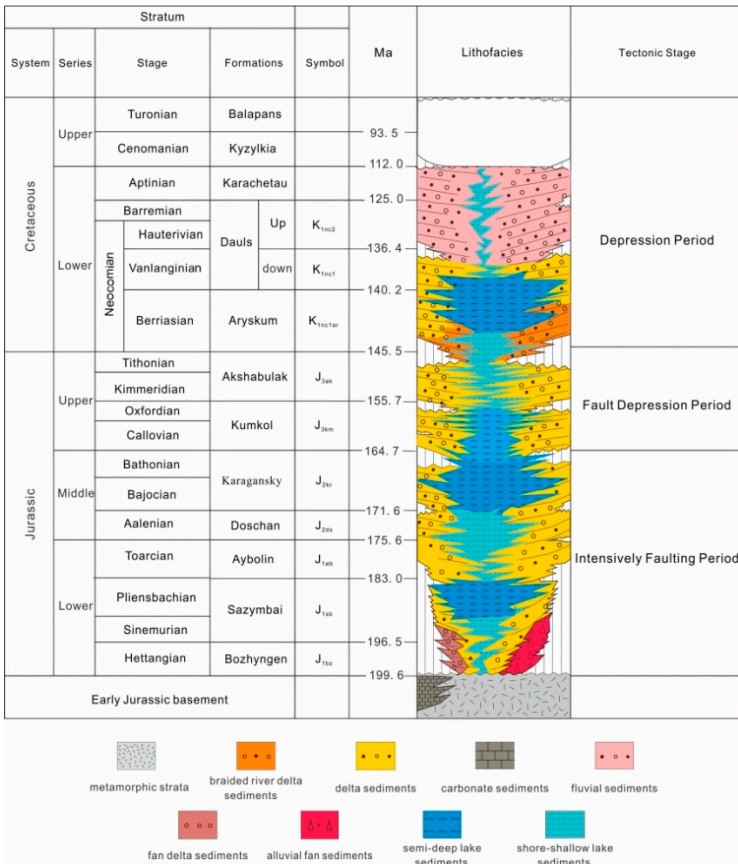

**Figure 2.** Stratigraphic column map of the South Turgay Basin.

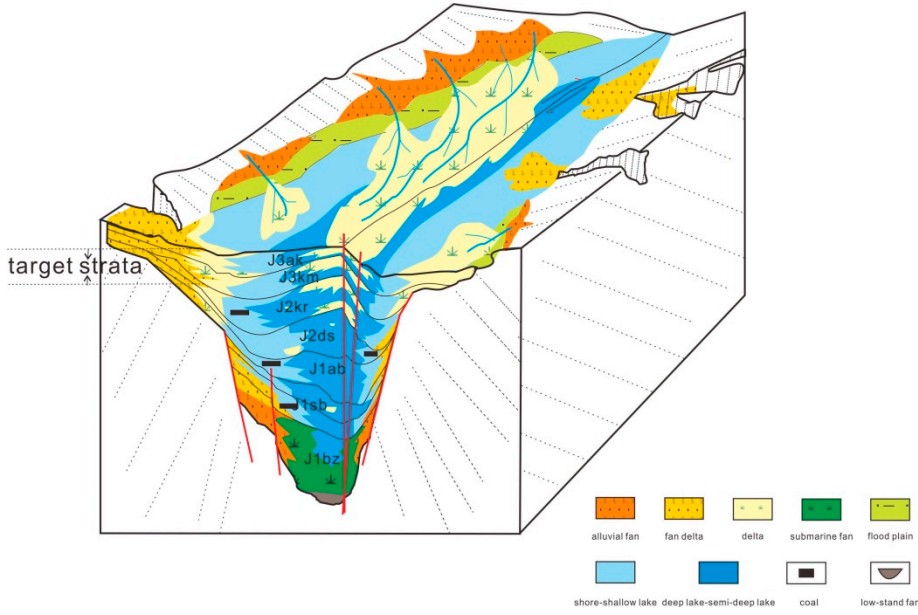

**Figure 3.** Map of sedimentary mode in the Jurassic, Aryskum Graben.

In the Cretaceous–Paleogene period, interbedded sandstone and mudstone of river–lake and alluvial origins were mainly developed. The Cretaceous stratum is mainly marine gray and terrestrial red sandstone, siltstone and mudstone; the Paleogene stratum is unconformity above the Cretaceous

with mudstone at the bottom; and the Neogene–Quaternary stratum is mainly sandy mudstone and unconformity above the Eocene [26].

The target strata studied in this paper are from the Sazymbai formation to the Akshabulak formation (Figure 2).

## 3. Methodology

The key component of this modeling study consists of the following three sections: 1) preparing the logging data, 2) training the SVM classification model, and 3) conducting the automatic identification model (Figure 4). The following subsections introduce the steps in detail.

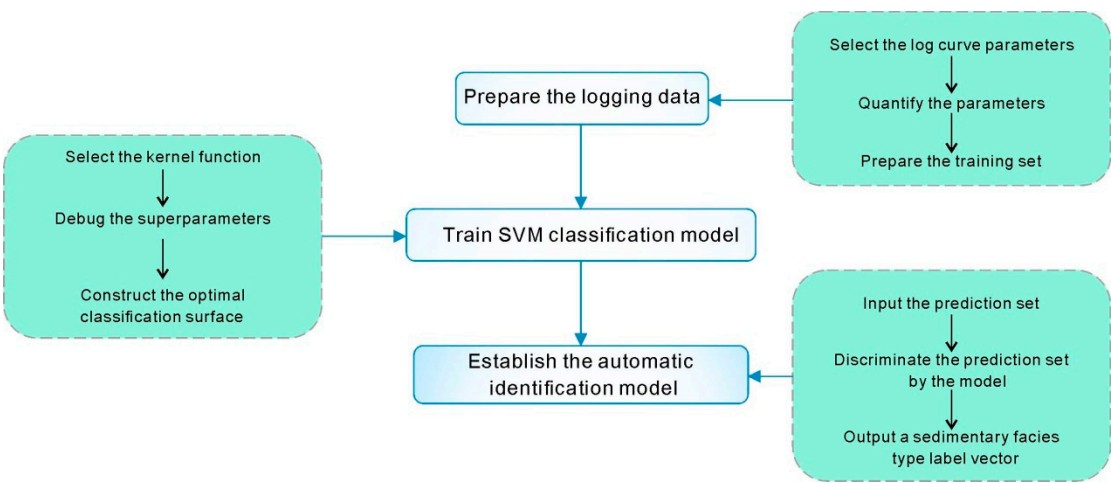

**Figure 4.** Conceptual framework for the modeling process [27,28].

### 3.1. Prepare the Logging Data

Based on the analysis of sedimentary characteristics of the target strata in the study area (Figure 3), three sedimentary subfacies, delta front, delta plain and shore-shallow lake, were selected as the label values for this classification task. When the label values were determined, the natural gamma-ray (GR) curve was chosen as the logging curve to extract the logging facies elements, and six logging facies elements were selected to quantitatively describe the shape of the logging curve. The curve was sampled at equal depth intervals from top to bottom to form an $n + 1$ logging data sequence. In this way, the depth sequence $d_0, d_1,..., d_n$ and its corresponding log amplitude sequence $a_0, a_1,..., a_n$ can be obtained. The thickness was recorded as $h_i = d_i - d_{i-1}$ ($i = 1, 2,..., n$). After obtaining the depth sequence and corresponding logging amplitude sequence, the corresponding parameters could be quantified.

(1) Variance

This property is expressed by the standard deviation of the curve:

$$\sigma = \sqrt{\frac{1}{n-1}\sum_{i=1}^{n}(a_i - \bar{a})^2} \tag{1}$$

The standard deviation of the curve actually reflects the sorting of the grain sizes. If the grain sizes of the sediment are sorted well and the physical property changes little, then the logging curve is an approximately straight line. The mean value of the logging data is mainly concentrated in the middle of the curve. Only the top and bottom data deviate from the mean value slightly, so the value of $\sigma$ is smaller. At this time, it corresponds to the box curve. On the contrary, if the separation of sediment grain sizes is poor, the physical properties change greatly. It shows certain fluctuations in logging curves and the shape of the curves may be a funnel, bell or finger. At this time, the logging data are relatively scattered, and a considerable section of the data points are deviate greatly from the

mean, which will inevitably lead to a larger value of $\sigma$. The calculation shows that the variance of large mudstone layers is the smallest in all sedimentary environments, which corresponds to the deposition of mudstone between rivers or deltas [29].

(2) Relative Center of Gravity

The relative center of gravity (RM) reflects the curve shape and can also explain the graded sequence of the sedimentary rock grains. It can be expressed as follows:

$$\mathrm{RM} = \frac{\sum_{i=1}^{n} i a_i}{n \sum_{i=1}^{n} a_i} \ (i = 1, 2, \ldots, n) \tag{2}$$

The grain sequence of sedimentary facies is different between positive and negative. Generally, the center of gravity of the positive grain order is lower than that of the reverse grain order, and the center of gravity of the reverse grain order is higher than that of the normal grain order. However, because the length of each section is different, the relative center of gravity is taken here to facilitate a comparison [30]. When RM > 0.5, the center of gravity is lower, and the curve is bell-shaped. When RM < 0.5, the curve is funnel-shaped, and the center of gravity is higher. When RM $\approx$ 0.5, the curve shape is a box type [11].

(3) Ratio Amplitude

Ratio amplitude refers to the ratio of the amplitude of a curve to its thickness. Different curve shapes have different ratios, which reflect lithological changes to a certain extent. By calculation, the maximum and minimum values are found. Taking the GR curve as an example, the ratio of minimum value to thickness is defined as the left ratio amplitude ($\lambda_{\mathrm{L}}$), and the ratio of maximum value to thickness is defined as the right ratio amplitude ($\lambda_{\mathrm{R}}$). Obviously, the larger the left ratio amplitude, the coarser the grain size of the sediment; the smaller the left ratio, the smaller the grain size of the sediment [31].

(4) Average Median

When a logging curve has only one curve, the average median of the curve (AM) is usually used to measure the amplitude difference of the logging curve [31], and it can be calculated by Equation (3):

$$\mathrm{AM} = 0.5(\bar{a} + ME) \tag{3}$$

The AM reflects the concentration degree of the primary amplitude values of the curve [11].

(5) Average Slope

The average slope of the curve is recorded as $\overline{K_p}$ and can be expressed as follows:

$$\overline{K_p} = \frac{\sum_{i=1}^{n} (a_i - \bar{a}) \cdot (d_i - \bar{d})}{\sum_{i=1}^{n} (a_i - \bar{a})^2} \tag{4}$$

The inclination angle of the average slope straight line can be obtained using $\overline{K_p}$, and is recorded as $T$ ($T = 57.2958^{\circ}\mathrm{arctg}\ \overline{K_p}$). Then, the shape of the curve is determined according to $T$ [31].

(6) Mutational Amplitude Difference

The mutational amplitude difference is recorded as $a_{\mathrm{m}}$, and can be expressed as follows:

$$a_m = a_{max} - a_{min} \tag{5}$$

In Equation (5), $a_{\mathrm{max}}$ represents the maximum amplitude at the mutation, and $a_{\mathrm{min}}$ represents the minimum amplitude at the mutation. In this study, the top mutation amplitude was recorded as $a_{\mathrm{m\text{-}T}}$, and the bottom mutation amplitude was recorded as $a_{\mathrm{m\text{-}B}}$. The contact relationship between top and bottom is determined by Equations (4) and (5). The nearer the average slope linear inclination $T$ is to $0^{\circ}$ or $180^{\circ}$, and the larger $a_{\mathrm{m}}$ is, the higher the degree of mutation is; otherwise, the degree of mutation is low [31].

After data preparation, the training set (Table 1) was input into the SVM classification model for model training.

**Table 1.** Sedimentary characteristics of log curves and corresponding subfacies in the study area (ARY-01).

| Characteristics Subfacies | $G$ | $S$ | $A_m$ | $A_{s\text{-}T}$ | $A_{s\text{-}B}$ | $\overline{K_p}$ | $\lambda_L$ | $\lambda_R$ |
|---|---|---|---|---|---|---|---|---|
| Delta front | 0.51 | 13.78 | 105.09 | 53.72 | 17.20 | 0.31 | 1.10 | 2.81 |
| Shore-shallow lake | 0.52 | 11.79 | 119.26 | 35.20 | 47.49 | 0.96 | 1.39 | 2.66 |
| Delta front | 0.55 | 19.27 | 110.43 | 20.59 | 43.87 | 0.93 | 1.06 | 1.97 |
| Delta plain | 0.47 | 32.33 | 109.83 | 83.92 | 114.18 | −0.23 | 0.77 | 2.70 |

### 3.2. SVM Classification Model

#### 3.2.1. SVM

The SVM is the core concept, and its accuracy is directly related to the accuracy of the classification results. Taking a classical binary classification task as an example, the basic principle of the SVM is introduced [32,33].

The mechanism of an SVM is to find an optimal classification hyperplane that meets the classification requirements, so that the hyperplane can maximize the blank areas on both sides of the hyperplane while ensuring classification accuracy. In theory, an SVM can achieve optimal classification of linear separable data.

The training sample set is given as $(x_i, y_i)$, $i = 1, 2, 3, \ldots , l$, $x \in R$, $y \in \{ \pm 1\}$, and the segmentation hyperplane is marked as $(w \cdot x) + b = 0$. To classify all samples correctly and obtain classification intervals, the following constraints must be satisfied:

$$y_i[(w \cdot x_i) + b] \geq 1 \quad i = 1, 2, 3, \ldots , l \tag{6}$$

The objective of optimization is to find a straight line ($w$ and $b$), to make the nearest point to the line as far as possible. The distance from point to line can be expressed as the following:

$$d = \frac{y_i[(w \cdot x_i) + b]}{\|w\|} \tag{7}$$

The optimization objective changes to Equation (8),

$$\underset{w,b}{\mathrm{argmax}}\{\frac{1}{\|w\|}\underset{i}{\min} y_i[(w \cdot x_i) + b]\} \tag{8}$$

Since $y_i[(w \cdot x_i) + b] \geq 1$, only $\underset{w,b}{\mathrm{argmax}}\frac{1}{\|w\|}$ should be considered. In addition, the solution of the maximum value of $\frac{1}{\|w\|}$ is equivalent to the solution of the minimum value of $\frac{1}{2}\|w\|^2$, so the objective function can be expressed as follows:

$$\begin{cases} \underset{w,b}{\min}\frac{1}{2}\|w\|^2 \\ s.t. \quad y_i[(w \cdot x_i) + b] \geq 1 \quad i = 1, 2, 3, \ldots , l \end{cases} \tag{9}$$

A Lagrange multiplier can be used to solve constrained optimization problems. The Lagrange relaxation method can be expressed as follows:

$$L(w, b, \propto) = \frac{1}{2}\|w\|^2 - \sum_{i=1}^{l} \propto_i \{y_i[(w \cdot x_i) + b] - 1\}, \propto_i \geq 0 \tag{10}$$

The solution to the constrained optimization problem is determined by the saddle point of the Lagrange function, and the solution to the optimization problem satisfies its partial derivative of $w$ and $b$ at the saddle point of 0. In addition, according to the dual property, the above optimal classification surface problem can be transformed into solving the maximum value of Equation (11) under the constraints of $\propto_i \geq 0$ and $\sum_{i=1}^{n} \propto_i y_i = 0$ $(i = 1, 2, 3, \ldots, n; j = 1, 2, 3, \ldots, n)$.

$$\sum_{i=1}^{n} \propto_i - \frac{1}{2} \sum_{i=1}^{n} \sum_{j=1}^{n} \propto_i \propto_j y_i y_j (x_i \cdot x_j) \tag{11}$$

The optimal solution of Equation (11) can be expressed as follows:

$$\propto^* = \left( \alpha_1^*, \alpha_2^*, \cdots, \alpha_l^* \right)^T \tag{12}$$

When the optimal solution is obtained, the optimal weight vector $w^*$ and the optimal bias $b^*$ are, respectively:

$$\begin{cases} w^* = \sum_{j=1}^{l} \alpha_j^* y_j x_j & j \in \{j \big| \alpha_j^* > 0\} \\ b^* = y_j - \sum_{j=1}^{l} y_j \alpha_j^* (x_j \cdot x_i) & j \in \{j \big| \alpha_j^* > 0\} \end{cases} \tag{13}$$

The optimal classification hyperplane can be obtained by Equation (13):

$$(w^* \cdot x) + b^* = 0 \tag{14}$$

The corresponding optimal classification surface function can be expressed as follows:

$$f(x) = sign\{(w \cdot x_i) + b\} = sign\left\{ \sum_{i=1}^{n} \alpha_i^* y_i (x_i \cdot x) + b^* \right\} \tag{15}$$

### 3.2.2. Kernel Function

The abovementioned principle is a method for calculating the linear separable time of a given sample. However, the given sample point is usually linearly inseparable in practical research. In the case of linear inseparability, the main idea of an SVM is to map the input sample points to their own higher-dimensional eigenvector space through the kernel function. Then, the optimal classification surface is constructed in the space. It can not only complete high-dimensional spatial operations in low-dimensional space but can also effectively avoid dimensional disasters [34] (Figure 5).

Some of the most commonly used kernel functions are as follows [35]:

(1) Radial basis function (RBF), which is defined by Equation (16),

$$K(x_i, x) = e^{-\gamma \cdot \|x_i - x\|^2} \tag{16}$$

(2) Polynomial function (PF), which is defined by Equation (17),

$$K(x_i, x) = \left( a x_i^T x + b \right)^c \tag{17}$$

(3) Sigmoidal function (SF), which is defined by Equation (18),

$$K(x_i, \ x) = \tanh\left(ax_i^T x + b\right) \tag{18}$$

In this study, the radial basis function (RBF), which is widely used at present, is selected for classification.

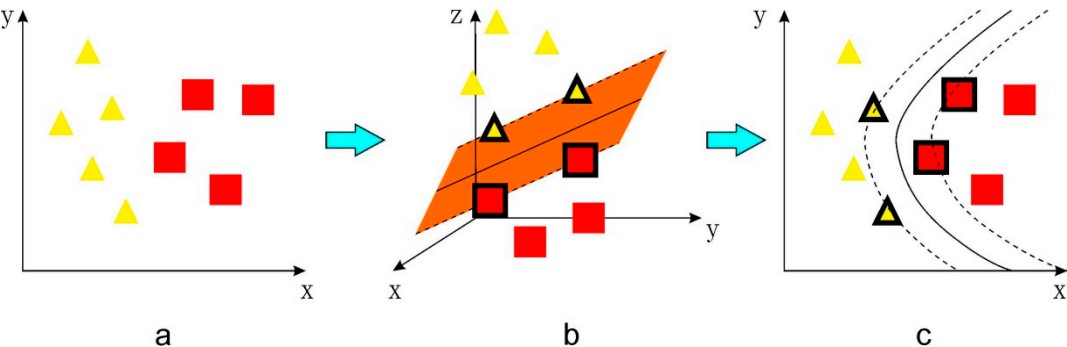

**Figure 5.** Creation of the boundary for a non-separable case. (**a**) Two linearly inseparable classes in two dimensions. (**b**) Projection onto a higher-dimensional space using the kernel function, where it is possible to separate the classes using a plane, with three support vectors indicated. (**c**) Projection back into two dimensions [35].

### 3.2.3. Optimal Classification Surface

After determining the kernel function, the sample points of 16 coring wells were debugged. The sample points of 13 coring wells (ARY-01–ARY-13) were used as the training set, and 3 coring wells (ARY14–ARY16) as the verification set. To prevent noise in the sample points, the concept of a soft margin is introduced in the training process. By introducing the relaxation factor $\xi_i$, Equation (6) is transformed into Equation (19),

$$y_i[(w \cdot x_i) + b] \geq 1 - \xi_i \quad i = 1, \ 2, \ 3, \ldots, l \tag{19}$$

The new objective function can be expressed as follows:

$$\min \frac{1}{2}\|w\|^2 + C\sum_{i=1}^{l} \xi_i \tag{20}$$

The parameter *C* is called the penalty parameter. When the value *C* is maximized, it means that there can be no mistakes in the classification. When the value *C* is minimized, it means that there is a higher fault tolerance rate in the classification. After introducing penalty parameters, the problem of solving the optimal classification surface is also transformed into seeking the maximum value of Equation (21):

$$\begin{cases} \sum_{i=1}^{n} \alpha_i - \frac{1}{2}\sum_{i=1}^{n}\sum_{j=1}^{n} \alpha_i\alpha_j \ y_iy_jK\left(\vec{x_i}, \ \vec{x_j}\right) \\ s.t. \ \sum_{i=1}^{n} \alpha_i \ y_i = 0 \\ 0 \leq \alpha_i \leq C \end{cases} \tag{21}$$

By solving Equation (21), $\alpha^* = \left(\alpha_1^*, \alpha_2^*, \cdots, \alpha_n^*\right)^T$ and the optimal bias $b^*$ can be found, and the corresponding optimal classification surface function is transformed into Equation (22):

$$f(x) = sign\{(w \cdot x_i) + b\} = sign\left\{\sum_{i=1}^{n} \alpha_i^* y_i K(x_i \cdot x) + b^*\right\} \tag{22}$$

In the process of model training, the method of "one against one" is used to realize the multi-classification task. The penalty parameter *C* and parameter $\gamma$ in the kernel function are determined by the K-fold cross-validation method. Finally, when the value of $\gamma$ is 1 and *C* is 10, the best result of subfacies identification is obtained.

## 4. Results and Discussion

### 4.1. Results

After the model training was completed, the model was used to determine the subfacies of test samples. Three coring wells (ARY14–ARY16) in the study area were selected as test sets to determine the accuracy of the classification model. The sedimentary subfacies of the three coring wells included three delta plains, five delta fronts, and four shore-shallow lakes, totaling 12 sedimentary subfacies.

Through the confusion matrix generated by the test set, and a comparison of the predicted results of the model output with the results identified by geologists (Figures 6–9), it can be seen that there are eight correct results and four incorrect results in the prediction model. In other words, the accuracy of the model is approximately 70%. Among the four incorrect results, two incorrect results are that shore-shallow lake is identified as delta plain (Figure 9). This is mainly because the sedimentary environments of the two are similar, which leads to the similarity of their quantitative index values. Therefore it is difficult to distinguish between them when there are only a few samples. The author believes that with the increase of the number of samples in the later stage, the accuracy of the model will be improved.

### 4.2. Discussion

This study compared the identification results between the SVM and the traditional method of manual identification (Figures 7–9). As shown in Figures 7–9, the SVM classification model successfully identified most of the subfacies. However, due to the paucity of logging data in the study area, its accuracy may not be very high. The author believes that with the increase of drilling quantity in the later exploration stage, the accuracy will increase. From this study, we can see that the SVM algorithm has unique advantages in small- and medium-sized sample statistical processing and is suitable for classification tasks with unbalanced numbers of samples. However, it should be noted that to realize the multi-classification task, the method of "one against one" was used. Any two sample sequences were combined to construct $C_N^2$ vector machines, the "vote" was adopted to classify and perform the identification of N types of depositional subfacies in the intervals [11]. It means that though this method is good, when there are N categories, the number of models is $\frac{N(N-1)}{2}$. Therefore, when using an SVM to realize multi-classification tasks, there should not be too many classification objectives. This is due to the fact that too many classification objectives will require a huge amount of computation and reduce the final classification accuracy.

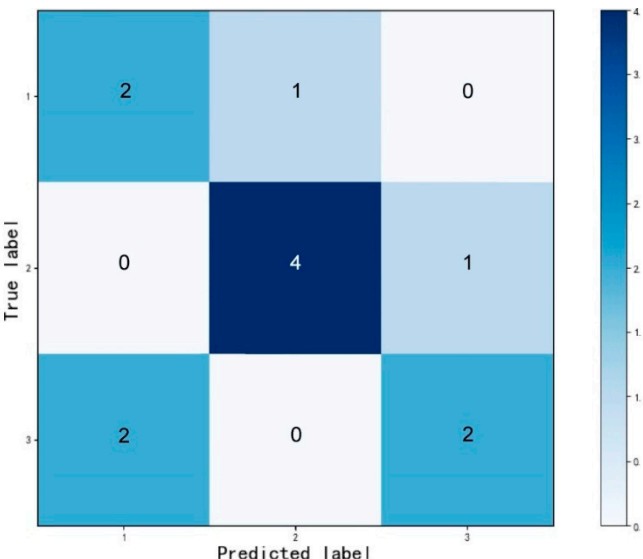

**Figure 6.** The confusion matrix of the support vector machine (SVM) classification model.

| Stratum | | | Depth (m) | GR | Sandstone | Comparison of predicted results | |
|---|---|---|---|---|---|---|---|
| System | Series | Formations | | 0 ——— 200 | 200 ——— 0 | Subfacies (specialist) | Subfacies (SVM) |
| Jurassic | Upper Jurassic | J3ak | 1400 — 1500 — | | | delta plain | delta plain |
| | | | | | | delta front | shore-shallow lake |
| | | J3km | 1600 — 1700 — | | | shore-shallow lake | shore-shallow lake |
| | | | | | | delta front | delta front |

Correct prediction     Wrong prediction

**Figure 7.** Comparison of the subfacies identification results by the SVM and geologists in well ARY14.

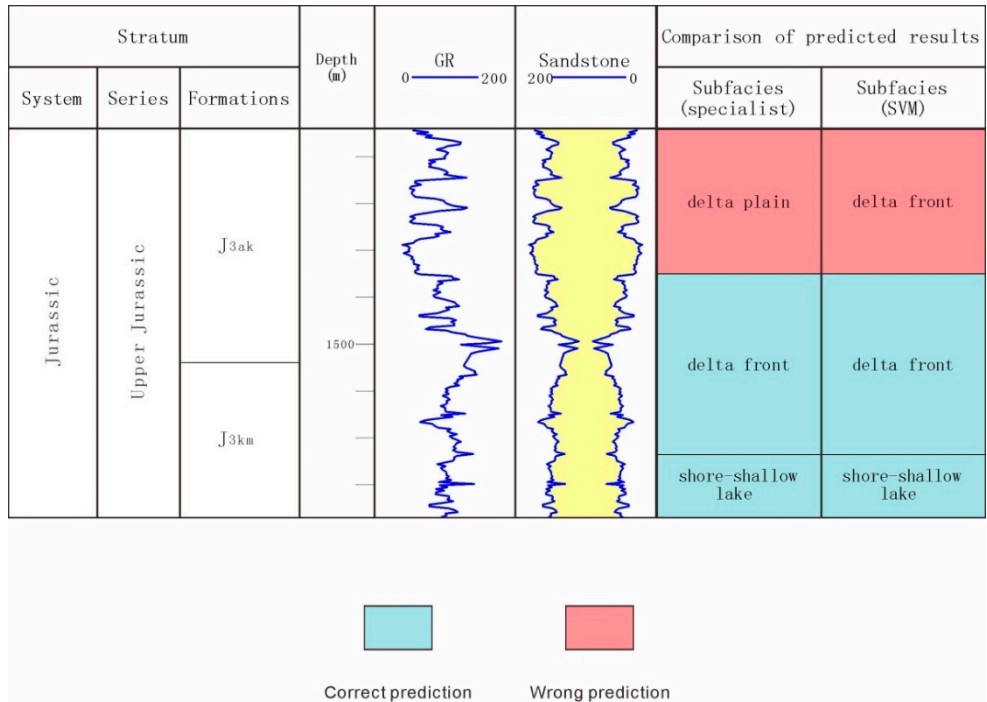

**Figure 8.** Comparison of the subfacies identification results by the SVM and geologists in well ARY15.

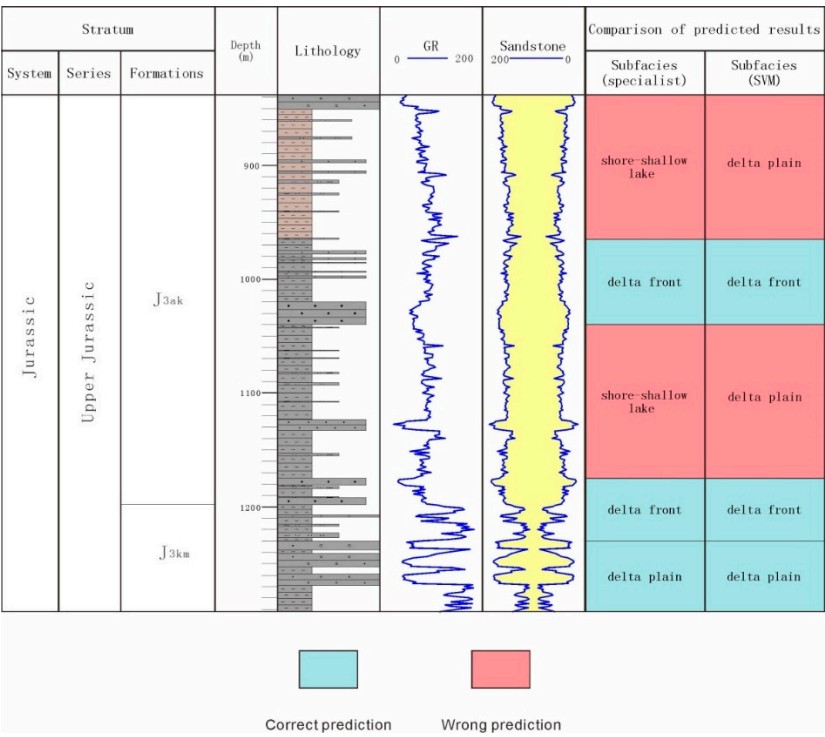

**Figure 9.** Comparison of the subfacies identification results by the SVM and geologists in well ARY16.

## 5. Conclusions

In this study, the GR curves of 16 coring wells in the Aryskum Graben in the South Turgay Basin were used, and the automatic identification of sedimentary facies in this area was completed using the SVM algorithm. The following conclusions and achievements have been obtained.

Firstly, the accurate and quantitative description of log curves is the foundation for the study. So, based on the GR curves of the 16 coring wells and previous research results, six quantitative indexes, including variance, relative gravity center, ratio amplitude, average median, average slope, and mutational amplitude difference, were selected to quantify the logging curves in the study area, and a description of the logging curves in the study area was realized.

Secondly, correct values of hyper-parameters and an appropriate kernel function are also the key factors in determining the accuracy of the SVM classification model. The training samples and radial basis function (RBF) were selected to map the input sample points into a higher-dimensional eigenvector space. Then, the hyper-parameters $\gamma$ and $C$ were debugged by a cross-validation method. When the value of $\gamma$ was 1 and $C$ was 10, the best result of subfacies identification was obtained.

Thirdly, the classification model was applied to the ARY14–ARY16 wells. The results show that the classification model can effectively accomplish the automatic identification task of sedimentary facies in the study area, and the accuracy is approximately 70%.

In summary, this study provides an innovative, accurate and efficient solution for identifying depositional subfacies. The methodology shown here will provide researchers with new ideas for sedimentary facies identification. Combined with the research results of this paper, our work can provide a theoretical basis for the fine division of microfacies in the next stage of the study. However, due to the paucity of logging data in the study area, its accuracy may not be very high. Moreover, if there are too many classification objectives, the computational effort will increase significantly and the final accuracy may not be as high as the results of this study. The future work to improve this method will focus on the optimization of the algorithm.

**Author Contributions:** Formal analysis, X.A.; methodology, B.S. and X.A.; resources, H.W.; supervision, H.W. and B.S.; writing—original draft, X.A.; writing—review & editing, X.A.

**Funding:** This research was funded by the National Key R&D Program of China (Grant No. 2017YFC1500604), the Preliminary Study on Earthquake Risk Assessment Model for Typical Engineering Structures (Grant No. 2019EEEV0103) and the Program for Innovative Research Team in China Earthquake Administration (Earthquake Disaster Simulation and Evaluation in mainland of China).

**Acknowledgments:** The authors thank Wang Hongyu of China University of Geosciences (Beijing) and Sun Baitao of Institute of Engineering Mechanics, CEA for their constructive help.

**Conflicts of Interest:** The authors declare no conflict of interest.

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
