# Peer review of "Automatic Identification of Sedimentary Facies Based on a Support Vector Machine in the Aryskum Graben, Kazakhstan"

_applsci, doi:10.3390/app9214489_

Round 1

Reviewer 1 Report

The originality, accuracy and completeness of the work are satisfactory. Typing errors of the manuscript should be corrected. Please see the attached file for your reference.

Author Response

Point 1: Typing errors of the manuscript should be corrected. Please see the attached file for your reference.

Response 1: Thank you for your comments. I have already corrected all my references in the format of this magazine. Please see the attachment.

Reviewer 2 Report

My recommendations are given below.

More citations are necessary to the section 2. Section 4 is inadequate. It should be extended in several times because the results and their interpretation are the "core" of each good academic paper. Moreover, there should be two sections, namely Results and Discussion. Results: your direct findings. Discussion: interpretation of your findings and putting them into the internationally-sounding context. Conclusions should list the main findings, not just declarative statements. The writing is clear, but requires some polishing. E.g., Line 133: the word "ended" is not good; heading 2.2: stratum -> strata. etc. etc. Please, polish everywhere! I strongly encourage the authors to format your submission, including the list of references, strictly according to the rules of the journal.

Author Response

Dear Reviewer,

Thank you for your comments and here is my responses:

Point 1: More citations are necessary to the section 2.

Response 1: More citations have been added in the manuscript ( [23-25] ). 

Point 2: Section 4 is inadequate. It should be extended in several times because the results and their interpretation are the "core" of each good academic paper. Moreover, there should be two sections, namely Results and Discussion. Results: your direct findings. Discussion: interpretation of your findings and putting them into the internationally-sounding context.

Response 2: Section 4 has been divided into two parts: results (4.1) and discussion (4.2). Here is part of the Discussion section:

"But it should be noted that to realize the multi-classification task, the method of “one against one” was used. Any two sample sequences were combined to construct  vector machines, the "vote" was adopted to classify and perform the identification of N types of depositional subfacies in the intervals [11]. It means that though this method is good, when there are N categories, the number of models is N*(N-1)/2. So, when using SVM to realize multi-classification tasks, the classification objectives should not be too many. Because too many classification objectives will generate huge amount of computation and reduce the final classification accuracy."

Point 3: Conclusions should list the main findings, not just declarative statements.

Response 3: Conclusions have been reedited according to your comments. 

Point 4:The writing is clear, but requires some polishing.

Response 4: The relevant words have been modified.Before I submit this article, it has been submitted to a professional organization for language polish. I will upload the certificate to you as attachment. If it doesn't meet your requirements, I will ask this magazine to polish the language later.

Point 5:I strongly encourage the authors to format your submission, including the list of references, strictly according to the rules of the journal. 

Response 5: The manuscript has been reedited according to your comments, especially in the References section. Here is part of the References section:

"1. Li, Y.H.; Wang, H.T.; Wang, M.C.; Lian, P.Q.; Duan, T.Z.; Ji, B.Y. Automatic identification of carbonate sedimentary facies based on PCA and KNN using logs. Well Logging Technol. 2017, 41, 57-63.

 2. Lakzaie, A., Ghasem-alaskari, M.K., et al. Reservoir Facies Modeling Using Intelligent Data Gathering in an Iranian Carbonate Field. SPE 121247, 2009: 1-4.

"

Round 2

Reviewer 2 Report

I'm very satisfied with this revision. I only ask to extend the section Results. Each section should consist of at least 2 paragraphs. So, please, describe your results in more detail.

Author Response

Dear reviewer,

Thank you for your affirmation of my manuscript.

Here is my response:

Point 1:I only ask to extend the section Results. Each section should consist of at least 2 paragraphs.

Response 1:I have added the section Results to two paragraphs. Here is part of the section Results:

"Through the confusion matrix generated by the test set, and comparison of the predicted results of the model output with the results identified by geologists (Figs. 6–9), it can be seen that there are eight correct results and four incorrect results in the prediction model. In other words, the accuracy of the model is approximately 70%. Among the four incorrect results, two incorrect results are that shore-shallow lake is identified as delta plain (Fig. 9). This is mainly because the sedimentary environment of the two is similar, which leads to the similarity of their quantitative index values. So it is difficult to distinguish them when there are only a few samples. The author believes that with the increase of the number of samples in the later stage, the accuracy of the model will be improved."